# Cell-to-Cell Transmission of HIV-1 and HIV-2 from Infected Macrophages and Dendritic Cells to CD4+ T Lymphocytes

**DOI:** 10.3390/v15051030

**Published:** 2023-04-22

**Authors:** Marta Calado, David Pires, Carolina Conceição, Rita Ferreira, Quirina Santos-Costa, Elsa Anes, José Miguel Azevedo-Pereira

**Affiliations:** 1Host-Pathogen Interactions Unit, Research Institute for Medicines, iMed-ULisboa, Faculty of Pharmacy, Universidade de Lisboa, Av. Prof. Gama Pinto, 1649-003 Lisboa, Portugaleanes@ff.ul.pt (E.A.); 2Center for Interdisciplinary Research in Health, Católica Medical School, Universidade Católica Portuguesa, Estrada Octávio Pato, 2635-631 Sintra, Portugal

**Keywords:** HIV, acute infection, transmission, cis-infection, trans-infection, macrophages, dendritic cells

## Abstract

Macrophages (Mø) and dendritic cells (DCs) are key players in human immunodeficiency virus (HIV) infection and pathogenesis. They are essential for the spread of HIV to CD4+ T lymphocytes (TCD4+) during acute infection. In addition, they constitute a persistently infected reservoir in which viral production is maintained for long periods of time during chronic infection. Defining how HIV interacts with these cells remains a critical area of research to elucidate the pathogenic mechanisms of acute spread and sustained chronic infection and transmission. To address this issue, we analyzed a panel of phenotypically distinct HIV-1 and HIV-2 primary isolates for the efficiency with which they are transferred from infected DCs or Mø to TCD4+. Our results show that infected Mø and DCs spread the virus to TCD4+ via cell-free viral particles in addition to other alternative pathways. We demonstrate that the production of infectious viral particles is induced by the co-culture of different cell populations, indicating that the contribution of cell signaling driven by cell-to-cell contact is a trigger for viral replication. The results obtained do not correlate with the phenotypic characteristics of the HIV isolates, namely their co-receptor usage, nor do we find significant differences between HIV-1 and HIV-2 in terms of cis- or trans-infection. The data presented here may help to further elucidate the cell-to-cell spread of HIV and its importance in HIV pathogenesis. Ultimately, this knowledge is critical for new therapeutic and vaccine approaches.

## 1. Introduction

The human immunodeficiency virus (HIV) is the causative agent of acquired immunodeficiency syndrome (AIDS). The syndrome was first recognized in 1981 and has since become one of the world’s major public health problems, particularly in developing countries.

HIV infects cells of the immune system, specifically CD4+ T lymphocytes, monocytes/macrophages, and dendritic cells. HIV transmission to a new host occurs through intact mucous membranes, eczematous or damaged skin or mucosa, and parenteral inoculation. The establishment of chronic infection requires that the transmitted virus productively infect its target cells: CD4+ T lymphocytes [1,2], monocytes [3], and dendritic cells [4,5,6].

During sexual transmission, HIV must cross the epithelial cell lining of the mucosa to gain access to the subepithelial layer. During this process, the virus interacts with cells present in the mucosa, such as the dendritic cells (including Langerhans cells) and macrophages.

Infection of CD4+ T lymphocytes (TCD4+) is a critical step in HIV transmission and appears to occur via a two-step mechanism [7]. The first is by trans-infection from dendritic cells (DCs) and macrophages (Mø). These cells, after being challenged with transmitted viral particles, could take up the incoming virus into non-degradative cytoplasmic compartments and transport it to regional lymph nodes, where it can be transferred to CD4+ T lymphocytes. The interaction of HIV with DCs is complex and multifaceted and strongly depends on the maturation state of the DC [8]. However, several reports have indicated that HIV binding to DCs appears to be mediated by the interaction of the viral envelope glycoprotein (gp120) with C-type lectin receptors, such as DC-specific ICAM3-grabbing non-integrin—DC-SIGN, Langerin, DC immunoreceptor—DCIR, and syndecan-3 [9,10,11,12]. 

The second mechanism is called cis-infection, which involves productive infection of DCs and Mø with de novo production of viral particles that infect surrounding CD4+ T lymphocytes. In this case, HIV infects target cells through the interaction between gp120 and the specific receptors present on the cell membrane: CD4 and a chemokine receptor such as CCR5, CXCR4, and CCR8, among others [13,14,15,16].

In both cis- and trans-infection, HIV particles are transferred from donor cells (DCs or Mø) to receptor cells (CD4+ T lymphocytes) by a cell-to-cell synapse-induced mechanism [17,18,19]. This mechanism mimics that observed during the immunological antigen presentation process, where CD4+ T lymphocytes must interact with antigen-presenting cells (APC) such as DCs and Mø. During this process, the coordinated recruitment of adhesion molecules and HIV receptors, together with the polarization of viral assembly and budding, enables an efficient strategy exploited by HIV and other enveloped viruses for intercellular transfer [20,21].

After these initial steps, HIV can be detected in regional lymphoid tissue one to two days after infection [22] and in regional lymph nodes within 5–6 days. By 10–14 days post-infection, HIV has disseminated and can be detected in various body compartments, including peripheral blood, lymphoid tissues, the genital tract, and the central nervous system [23]. In these compartments, HIV either establishes a productive infection with the production of new virions or establishes a lifelong cellular reservoir as a result of its ability to integrate proviral DNA into the host cell chromosome. In these cell populations, the HIV genome is maintained in a latent state (i.e., without the production of viral particles) that is neither detected by the host immune response nor affected by antiretroviral drugs, thus contributing to long-lasting infection and the failure of curative therapy of HIV infection.

Although CD4+ T lymphocytes are the primary targets of HIV infection and the most extensively studied latent reservoir [24,25], other cells, such as monocytes, macrophages, dendritic cells, and microglial cells can also maintain HIV in a latent state [26,27]. By definition, once infected, these cells harbor an integrated, fully competent provirus that could be transcribed once the cell is activated with the production of new virions [28]. All mechanisms of HIV transmission and spread are determined by viral phenotype, such as the ability to use the CCR5 chemokine receptor together with CD4 to enter cells (R5 strains), a higher Env protein content/virion, improved dendritic cell interaction, and resistance to interferon [29,30,31].

Here, we analyzed the efficiency with which HIV-1 and HIV-2 primary isolates are able to interact with Mø, DCs, and CD4+ T lymphocytes in either cis- or trans-infection. Our results show that infected Mø and DCs spread the virus to TCD4+ via cell-free viral particles in addition to other alternative pathways. We demonstrate that the production of infectious viral particles is induced by the co-culture of different cell populations, indicating that the contribution of cell signaling driven by cell-to-cell contact is a trigger for viral replication. These results may help to further elucidate the cell-to-cell spread of HIV and its importance in HIV pathogenesis. Ultimately, this knowledge is critical for new therapeutic and vaccine approaches.

## 2. Materials and Methods

### 2.1. Cells Isolation and Culture

Peripheral blood mononuclear cells (PBMC) were obtained from a mixture of 3 buffy coats of healthy donors provided by the National Blood and Transplantation Institute (Instituto Português do Sangue e da Transplantação, Lisbon, Portugal). Briefly, PBMCs were first isolated by density gradient centrifugation using Ficoll–Paque Plus (GE Healthcare, Chicago, IL, USA). The PBMC fraction was incubated with anti-CD14 magnetic beads (Miltenyi Biotech), and the CD14+ monocytes were isolated using magnetic-activated cell separation (MACS) columns. Monocyte differentiation to Mø was induced by allowing them to adhere to 24-well plates at 3 × 10^5^ cells per well, for 2 h at 37 °C, 5% CO_2_, in an RPMI-1640 medium, GlutaMAX supplement, HEPES (Gibco, Billings, MT, USA). Following adherence, the medium was supplemented to achieve a final concentration of 10% (*v*/*v*) fetal bovine serum (FBS) (Hyclone, GE Healthcare), 1 mM of sodium pyruvate (Hyclone, GE Healthcare), 0.1% β-mercaptoethanol (Gibco), and 20 ng/mL of the recombinant human M-CSF (BioLegend). The cell culture medium was renewed every three to four days until day seven of differentiation.

To obtain immature monocyte-derived DC (imDCs), monocytes were cultured in 24-well plates at 4 × 10^5^ cells per well in an RPMI-1640 medium, GlutaMAX supplement, HEPES (Gibco) supplemented with 10% (*v*/*v*) fetal bovine serum (FBS) (Hyclone, GE Healthcare), 1 mM of sodium pyruvate (Hyclone, GE Healthcare), 0.1% β-mercaptoethanol (Gibco), 10 ng/mL of recombinant human GM-CSF (BioLegend) and 20 ng/mL of IL-4. The maturation of imDCs was induced by culturing them with the described cytokines supplemented with interferon-gamma (IFN-γ).

Autologous CD4+ T lymphocytes were obtained from healthy donors according to the isolation protocol described above. Negative selection of the CD4+ T lymphocytes was performed using CD4+ T Cell isolation kit II (Miltenyi Biotech, Gaithersburg, MD, USA). Isolated lymphocytes were stimulated with 3 µg/mL of PHA-L for three days and further cultured in a 75 cm^2^ flask at 2 × 10^6^ cells per mL in an RPMI-1640 medium (Hyclone, GE Healthcare) supplemented with 15% (*v*/*v*) FBS (Hyclone, GE Healthcare), 2 mM L-glutamine (Gibco), 50 µg/mL gentamicin, 2.5 µg/mL amphotericin B, and 20 UI/mL of human recombinant interleukin-2 (BioLegend, San Diego, CA, USA).

### 2.2. HIV Isolates

All primary HIV-1 and HIV-2 were isolated by coculturing PBMCs obtained from infected patients with PBMCs from uninfected individuals as described [14]. After isolation, viral stocks were established from low-passaged supernatants of infected PBMC and stored at −80 °C until further use.

Viral replication was assessed by reverse transcriptase activity in culture supernatants by an enzyme-linked immunosorbent assay (Lenti-RT kit, Caviditech, Uppsala, Sweden). 

The 50% tissue culture infectious dose (TCID_50_) of the viruses was determined in a one-round viral infectivity assay using a luciferase reporter gene assay in TZM-bl cells and calculated using the statistical method of Reed and Muench [32].

Strains were selected from the viral library of the Host–Pathogen Interaction Unit (HPIU) of iMed.ULisboa, [14], according to the following criteria: (i) co-receptor usage profile (CCR5 vs. CXCR4), and (ii) clinical stage of infection (symptomatic vs. asymptomatic). 

### 2.3. Cis-Infection of CD4^+^ T Lymphocytes, Mø, and DCs

CD4+ T-lymphocytes, Mø, imDCs, and mDCs were infected with 1000 TCID_50_/mL of different HIV-1 and HIV-2 primary isolates or left uninfected as controls. In cis-infection, viruses were added and incubated for 3 h in the presence of 3 µg/mL of polybrene (Sigma–Aldrich, MO, USA). Cells were then washed with PBS to remove unadsorbed virus particles and cultured in an appropriate medium (500 μL/well). Culture supernatants were collected at time 0 (immediately after inoculum removal and cell washing), day 3, 6, 9, and 12, and quantified by RT concentration until day 12 post-infection [14].

### 2.4. HIV-1 and HIV-2 LTR Genomic Region Amplification

To detect HIV proviral DNA integration, DNA was extracted from exposed cells at day 12 post-infection and used in a nested PCR protocol targeting the viral LTR. The first round of PCR amplification was performed using an *Alu*-specific sense primer in combination with a *gag* antisense (HIV-1 or HIV-2) specific primer. The PCR products were subjected to a second PCR assay targeting the R/U5 region of the LTR (HIV-1 or HIV-2). As a negative control, we used chromosomal DNA extracted from uninfected cells. The nucleotide sequences and target sequence of each primer are shown in Table 1 and Table 2.

### 2.5. Trans-Infection of CD4^+^ T Lymphocytes by HIV-Infected Mø and DCs

Peripheral blood mononuclear cells were obtained from healthy donors as described [14]. Mø, imDCs, mDCs, and TCD4+ were obtained as described above.

Mø, imDCs, and mDCs were infected with different primary HIV isolates for 60 min at 37 °C or left uninfected as controls. The cells were then washed with PBS to remove unadsorbed viral particles and co-cultured with autologous activated TCD4+ at donor:target cell ratios of 1:1 and 1:4. Since Mø, imDCs, and mDCs are cultured adherent to plastic in contrast to TCD4+, we were able to separate the donor from the target cells 15 h after infection, leaving only the target cells (non-adherent) in culture. As a control, HIV-infected DCs or Mø were co-cultured with TCD4+ separated by a transwell (Transwell with 0.4 µm pore polyester membrane cell culture inserts, Corning) to prevent cell–cell contact.

The efficiency with which HIV was transferred from exposed donor cells to target cells was assessed by analyzing the production of new progeny virions by target cells (TCD4+) by measuring the RT concentration. In parallel, we also analyzed the efficiency of cis-infection of Mø imDCs, and mDCs by measuring the RT concentration in culture supernatants.

### 2.6. Statistical Analysis

Each experiment was performed in triplicate and independently repeated three times. The statistical analysis was performed using SPSS software version 29.0.0.0 (SPSS Inc., Chicago, IL, USA). The univariate analysis was tested using χ2 and 2-tailed Fisher’s exact test in case of a small sample size. Statistical significance was assumed when *p* < 0.05.

## 3. Results

### 3.1. CD4 + T Lymphocytes, Macrophages, Immature, and Mature DC Cis-Infection by HIV

First, we selected a set of 14 primary isolates from the viral library of HPIU of the iMed.Ulisboa [14] (Table 3 and Table 4), of which seven were HIV-1 strains, and seven were HIV-2 strains. The selection of these primary isolates was based on the following criteria: (i) they were obtained from individuals at different stages of infection; (ii) with different co-receptor usage profiles; and (iii) with minimal in vitro passages to avoid the selection of HIV variants due to culture conditions.

We then analyzed the efficiency of cis-infection of TCD4+, Mø, and DCs by HIV-1 and HIV-2 primary isolates. We expected to observe different capacities to infect Mø and DCs among different isolates and aimed to correlate these with viral biotypes and patient clinical and immunological data.

As shown in Figure 1, Mø infection by HIV-2 isolates revealed that four isolates (UCFL2072, UCFL2073, ALI, and UCFL2074) were able to perform productive infection (assessed by RT concentration in culture supernatants), while the other three (UCFL2017, UCFL2032, and UCFL2037) did not show quantifiable productive infection during the 12 days of infection. Regarding HIV-2 infection of imDCs and mDCs, only two isolates showed de novo virus production: UCFL2073 and ALI.

For HIV-1, all isolates except UCFL1029 were able to productively infect TCD4+, while in Mø, only one isolate, UCFL1032, showed productive infection on days 6 and 9 of infection (Figure 2).

In imDCs infected with HIV-1, viral production was detected for isolate UCFL1032 at days 9 and 12. Regarding isolates UCFL1014 and UCFL1028, despite the concentration of RT being below 10 pg/mL, we found the results interesting because we observed an increase of 2 to 5-fold between days 6 and 12, which leads us to believe that imDC is probably susceptible to infection by these strains. This is reinforced by the fact that, for these two isolates, the RT concentrations were generally low in all cell populations compared to the other isolates. In contrast, in mDCs, no isolates revealed increased levels of RT concentration.

Statistical analysis of the data obtained from triplicates of three independent experiments is summarized in Table 5 and indicates that differential usage of CCR5, CXCR4, or CCR8 by HIV-1 and HIV-2 is not correlated (data not shown but always *p* > 0.05) with productive infection of Mø, imDCs, mDCs, and TCD4+. *p*-values ranged from 0.062 to 1.311.

### 3.2. HIV-1 and HIV-2 Integration into Host Cell DNA

In the group of HIV isolates where no productive infection was detected, we further analyzed whether the virus was able to enter the cell, reverse transcribe its genome, and integrate the proviral DNA into the host cell chromosomes. To do this, we performed a nested PCR targeting the cell genome (*Alu* regions) and the LTR of the integrated provirus (Figure 3). 

The results of this assay, summarized in Table 6 and Table 7, show that all HIV-1 and HIV-2 isolates are positive for PCR amplification, except for imDC and mDC exposed to HIV-2UCFL2037. Overall, the results indicate that the vast majority of viruses were able to enter, reverse transcribe, and integrate their genome into host cell DNA. The absence of de novo viral production, as indicated by the absence of viral-associated RT in the culture supernatants, suggests that in these cases, the block in the replicative cycle occurred after integration and is not related to the use of different co-receptors (a situation that could ultimately hinder viral entry).

### 3.3. Trans-Infection of CD4+ T Lymphocytes by HIV-Infected Mø and DCs

The analysis of the results obtained in cis infection assay demonstrates that HIV-2 isolates were a more heterogeneous group, with different characteristics regarding their ability to infect Mø, imDCs, and mDCs. This observation, combined with the high incidence of HIV-2 infection in Europe and a small number of studies compared to HIV-1 infection, led us to select four primary HIV-2 isolates to evaluate the efficiency of transmission from Mø, imDCs, and mDCs to CD4+ T lymphocytes, using the HIV-1Ba-L strain (UCFL1034) as a control. Based on this rationale, we included two isolates that showed productive infection in Mø, imDCs, and mDCs (ALI and UCFL2073) and two others (UCFL2017 and UCFL2032) that, although they did not show quantifiable productive infection, were able to enter, reverse transcribe, and integrate the proviral DNA (Table 8).

Regarding the efficiency of HIV-2 spread from infected Mø and DCs to autologous CD4+ T lymphocytes by trans-infection (assessed by productive infection of the latter), we observed that when Mø were used as donor cells, two viral isolates, UCFL2017 and UCFL2073, were transferred and productively infected CD4+ T lymphocytes. This was observed in three assay conditions: (i) cell–cell contact, allowing donor and target cells to be in co-culture throughout the experiment; (ii) donor and target cells separated by a transwell to prevent cell–cell contact; and (iii) donor and target cells separated 15 h after co-culture. The rationale for these three different culture conditions was to try to mimic some of the in vivo conditions under which donor/infected and target/uninfected cells might interact: no contact between cells, transient contact, or prolonged contact. 

Interestingly, the UCFL2017 isolate was unable to replicate in Mø, imDCs, and mDCs when these cells were directly exposed to the virus (cis-infection experiments). This, together with the results obtained in the donor–target cell co-culture experiments, suggests that viral transmission is not only dependent on exposure to infectious viral particles, as demonstrated in the condition where donor and target cells were separated by a transwell but are influenced by contact between target cells and infected donor cells (cell-to-cell transmission). The same set of results was observed for the control HIV-1 strain, UCFL1034.

Remarkably, we were also able to detect the presence of RT for the viral isolate UCFL2032 when the donor and target cells were maintained in contact, although no viral replication was detected in cis-infection, suggesting a mechanism involving cell-to-cell interaction required for efficient viral transmission between these cells.

For HIV-2ALI trans-infection of imDCs, viral production was detected under all conditions, as well as for HIV-1 UCFL1034. However, UCFL2073 replication occurred only when donor and target cells were in contact or separated by a transwell, suggesting that viral particle production is likely induced by the co-culture of donor and target cells. Similar to the results of Mø, UCFL2017 could be efficiently transferred to target cells both when donor and target cells were in contact and when they were separated 15 h after the beginning of coinfection.

Finally, when trans-infections were performed using mDCs as donor cells, we verified that there was viral production in the three conditions for isolate UCFL2073; in contrast, for isolates UCFL2017 and UCFL2032, we detected the presence of RT when donor and target cells were in contact and when they were separated by a transwell. Since no viral production was observed during the cis-infection of mDCs with these strains, our results suggest that the production of infectious viral particles is induced by the co-culture of mDCs with CD4^+^ T lymphocytes. Only one HIV-2 isolates, ALI, and the HIV-1 control, UCFL1034, showed productive infection both in cell-to-cell contact and when the cells were separated 15 h after infection. 

The overall results of trans-infection of CD4+ T lymphocytes by Mø, imDCs, and mDCs infected with HIV-1 and HIV-2 isolates are summarized in Table 9. From these, we can see that in several cases, an increase in virus transmission was observed when we used a 1:4 ratio, especially when donor and target cells were separated by a transwell. However, this is not a clear and general pattern and appears to be a strain-specific effect.

## 4. Discussion

Mø and DCs, together with CD4+ T cells, are the major cell targets of HIV infection in vivo and clearly play important and well-established roles in multiple aspects of HIV pathogenesis. They act in vivo as vehicles for the spread of HIV, but they also act as cellular reservoirs for the virus, representing a major obstacle to viral eradication [33]. Therefore, studies of the interactions between HIV-infected Mø and DCs with CD4+ T lymphocytes are important to elucidate the mechanisms underlying the interactions leading to viral infection of these cells. The use of clinically relevant HIV-1 and HIV-2 isolates is essential for such studies. In the experiments included in this report, we used viral isolates that were minimally passaged in PBMCs to reduce the selection of mutants that are better adapted to in vitro conditions and potentially less able to replicate in vivo events. 

In this report, we used Mø, immature and mature DCs, and analyzed the efficiency of cis- and trans-infection by distinct HIV-1 and HIV-2 clinical isolates. We investigate the ability of these cells to be directly infected and/or to transfer infectious virions to T-CD4+ lymphocytes. The existence of an extensive viral library in our Research Unit allows us to use a number of viral isolates with unique characteristics in terms of viral biotypes [14]. Our results reinforce the notion that Mø and DC infection play a particularly important role in contributing to HIV spread to CD4+ T lymphocytes and also as persistently infected cellular reservoirs.

The analysis of the efficiency of cis-infection of CD4^+^ T lymphocytes, Mø, and DCs by our panel of HIV-1 and HIV-2 primary isolates led us to conclude that the ability to productively infect Mø and DCs (i.e., to complete the viral replication cycle, allowing de novo production of viral particles) is independent of co-receptor usage (namely, CCR5, CXCR4, or CCR8). In general, the co-receptor usage profile of a given strain is used to define its cellular tropism [13]; for example, the ability to use the CCR5 co-receptor (R5 strains) is usually considered a prerequisite for Mø infection (M-tropic strains). Despite this association, several studies have shown that some R5 isolates are unable to infect Mø [34,35,36,37,38,39,40,41], and conversely, some CXCR4-using strains (X4) are M-tropic [15,39,42]. These observations led to a redefinition of the HIV tropism classification (reviewed in [43]). Infection of DCs can also occur independently of the HIV co-receptor usage profile [44,45,46]. 

In this study, we further show that productive viral infection of Mø and mature dendritic cells (mDCs) is more frequently observed in cells inoculated with HIV-2 isolates than with HIV-1. To reinforce this notion, none of the HIV-1 isolates were able to replicate in mDCs, and only one (out of a total of seven isolates tested) was replicated in Mø. Our group and others have shown that HIV-1 and HIV-2 infections have different characteristics. The lower virulence and transmissibility observed in vivo or the ability to infect a broader range of cells in vitro, dictated by entry pathways mediated by different cell receptors, are clear examples of the different interactions that HIV-1 and HIV-2 establish with target cells and the human host [14,47,48,49,50,51,52,53,54,55,56,57].

Interestingly, despite the differences observed in productive infection, early events of the HIV replication cycle (i.e., adsorption, fusion, reverse transcription, and integration) occur even in those cases where de novo viral production was not detected (except for the UCFL1032 isolate). These data suggest that the abortive replication cycle is due to a blockage at some event after the integration of the provirus into the target cell genome. Alternatively, the lack of viral detection in supernatants of exposed cells may reflect a low-level viral replication in Mø and DCs. This lower viral replication is certainly strain-dependent and could be the result of the production of poorly infectious viral particles that accumulate in intracellular structures called virus-containing compartments (VCCs) [58,59,60]. 

Regarding HIV transfer from infected Mø and DCs to uninfected CD4+ T lymphocytes, our data show that virus spread is not only dependent on the exposure to cell-free virus produced by infected cells but is also influenced by the co-culture between target cells and infected donor cells. These culture conditions can induce the formation of a specialized structure, called the virological synapse, between the infected donor cell and the target cell [20]. The virological synapse facilitates the spread of HIV between cells allowing the virus to overcome some of the barriers present during cell-free viral infection [61,62]. For example, infection by cell-free viral particles is mechanistically more difficult because, during cis-infection, the virus must encounter a target cell by diffusion in the culture fluid phase. In fact, some technical improvements, such as spinoculation, were designed to increase the number of viral particles deposited on the target cell membrane, thereby facilitating viral adsorption to the CD4 receptor [63].

In contrast, the cell-to-cell spread of HIV is much more efficient, either because donor/infected and receptor/susceptible cells are in close contact avoiding this long-distance diffusion; or because contact between donor and receptor cells allows HIV to overcome cell barriers, thus leading to an increased viral replication in the latter. The cell–cell contact during co-culture between Mø or DCs with CD4+ T-lymphocytes could favor the concentration of viral particles in a limited area of the donor cell membrane locally, increasing the multiplicity of infection [64]. In addition, the formation of the virological synapse not only concentrates virions in a restricted area of contact between the donor and target cell but also: (i) triggers the remodeling of filamentous actin, facilitating the entry of the cell-free virus into the target cell [17,65]; (ii) induces recruitment of HIV cell receptors to the cell–cell junction [66]; (iii) allows the virus to evade the negative effect of some antiviral cellular factors, such as tetherin and TRIM5α [64,67,68]; and (iv) cell signaling driven by cell–cell contact creates a more favorable cellular environment for HIV replication [69]. The cell-to-cell contact observed in co-culture between infected immature DCs and CD4+ T cells may even allow viral replication to occur in less susceptible cells, such as resting T-lymphocytes [70].

In conclusion, our results show that (i) HIV-1 and HIV-2 have distinct abilities to infect Mø and DCs; (ii) the infection of these cells is independent of the co-receptor usage profile of the infecting virus; (iii) cell-to-cell contact facilitates HIV transfer from Mø or DCs to T-lymphocytes, regardless of viral phenotype; and (iv) in some cases, the cis-infection of Mø and DCs leads to an abortive replication cycle after proviral genome integration. This in vitro observation may provide a mechanism by which HIV establishes latently infected cells, which is particularly important for viral pathogenesis [71].

## Figures and Tables

**Figure 1 viruses-15-01030-f001:**
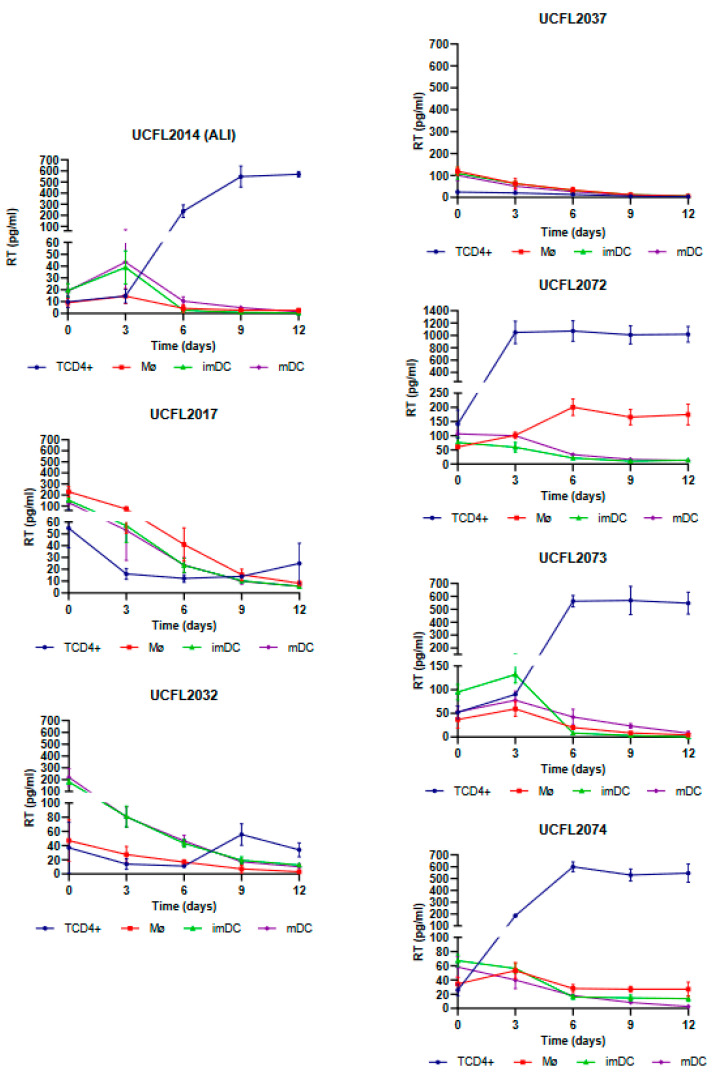
Cis-infection of TCD4+, Mø, imDCs, and mDCs by HIV-2 primary isolates. For better visualization, the data obtained for each virus are presented as individual line graphs and identified by the virus name above the graph. Viral replication was assessed by quantifying the RT concentration (pg/mL) in culture supernatants over the 12 days following infection. Productive infection was considered positive when RT concentration was equal to or above 10 pg/mL. Values depict the mean RT concentration of three independent assays measured in triplicate, while error bars depict the standard deviation (SD).

**Figure 2 viruses-15-01030-f002:**
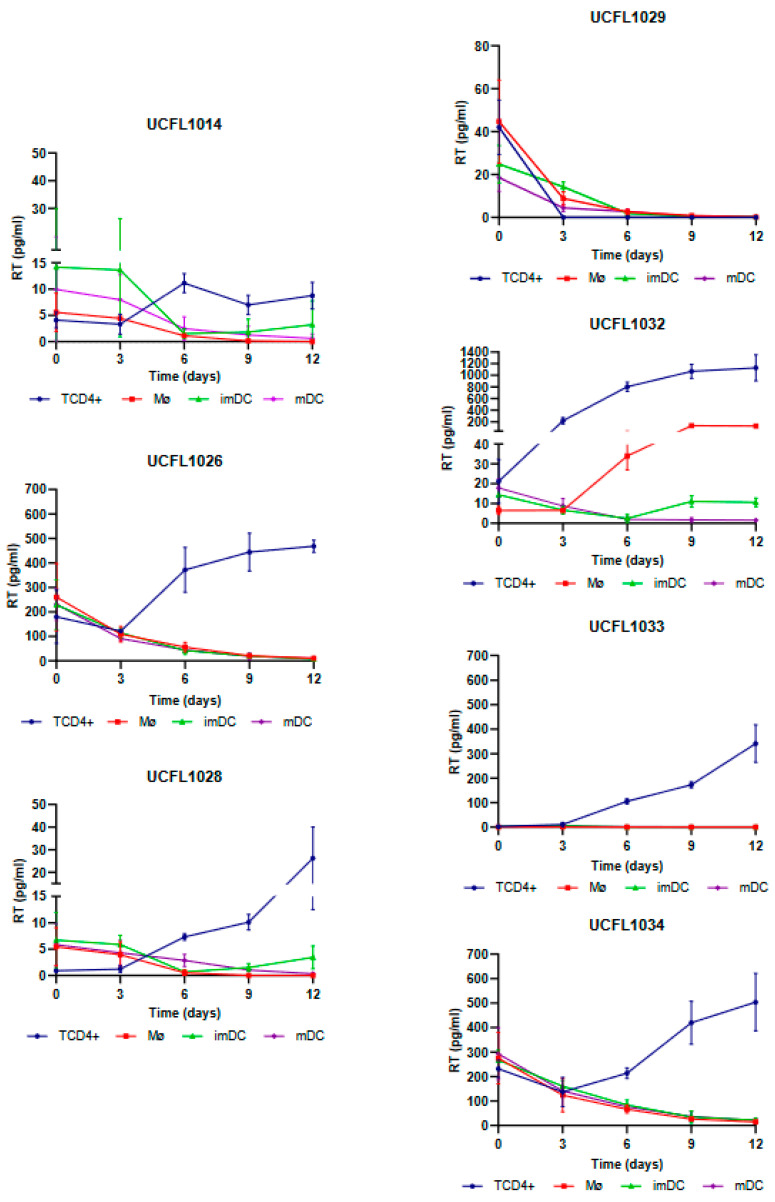
Cis-infection of TCD4+, Mø, imDCs, and mDCs by HIV-1 primary isolates. For better visualization, the data obtained for each virus are presented as individual line graphs and identified by the virus name above the graph. Viral replication was assessed by quantifying the RT concentration (pg/mL) in culture supernatants over the 12 days following infection. Productive infection was considered positive when RT concentration was equal to or above 10 pg/mL. Values depict the mean RT concentration of three independent assays measured in triplicate, while error bars depict the standard deviation (SD).

**Figure 3 viruses-15-01030-f003:**
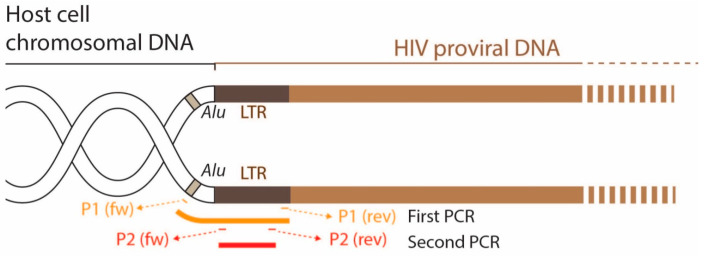
Diagram showing the position of the primers used in the nested PCR. The aim was to verify whether reverse transcription and integration of the proviral DNA into the cell genome occurred in cells where no virus production was detected.

**Table 1 viruses-15-01030-t001:** Forward and reverse primer nucleotide sequences for the polymerase chain reaction amplification of the HIV-1 LTR region.

Primers	Target Region	Nucleotide Position (bp)	Sequence (5′ to 3′)
**Alu (F)**	Alu	-	tcc cag cta ctg ggg agg ctg agg
**1LTR1 (R)**	LTR	516–540	agg caa gct tta ttg agg ctt aag c
**1LTR2 (F)**	LTR	45–71	ctg tgg atc tac cac aca caa ggc tac
**1LTR3 (R)**	LTR	411–436	gct gct tat atg tag cat ctg agg gc

**Table 2 viruses-15-01030-t002:** Forward and reverse primer nucleotide sequences for the polymerase chain reaction amplification of the HIV-2 LTR region.

Primers	Target Region	Nucleotide Position (bp)	Sequence (5′ to 3′)
**Alu**	Alu	-	tcc cag cta ctg ggg agg ctg agg
**2LTR1**	LTR	513–537	gcc tct ccg cag agc gac tga ata c
**2LTR2**	LTR	75–100	cca gat tgg cag gat tac acc tca gg
**2LTR3**	LTR	363–384	gcc atg tta gaa ggc ctc ttg c

**Table 3 viruses-15-01030-t003:** Clinical, immunological, and virological characteristics of HIV-1 infected patients and the biotype of the HIV-1 isolates obtained from them.

Isolate	Clinical Stage	CD4+ Lymphocytes Count (Cells/μL)	Viral Load (RNA Copies/mL)	Biotype ^a^
**UCFL1014**	Symptomatic	305	67,508	R5X4
**UCFL1026**	Symptomatic	278	178,364	R5X4
**UCFL1028**	Symptomatic	218	237,482	R5X4
**UCFL1029**	Symptomatic	1455	<500	R5X4
**UCFL1032**	ND	ND	ND	X4
**UCFL1033**	ND	ND	ND	R5
**UCFL1034 ^b^**	Symptomatic	ND	ND	R5

^a^ Biotype according to co-receptor usage; R5, usage of CCR5; X4, usage of CXCR4; R5X4, usage of CCR5 and CXCR4; ^b^ Reference strain Ba-L; ND, not determined.

**Table 4 viruses-15-01030-t004:** Clinical, immunological, and virological characteristics of HIV-2 infected patients and the biotype of the HIV-2 isolates obtained from them.

Isolate	Clinical Stage	CD4+ Lymphocytes Count (Cells/μL)	Viral Load (RNA Copies/mL)	Biotype ^a^
**ALI**	Symptomatic	491	ND	R5
**UCFL2017**	Symptomatic	50	23,454	R5X4R8
**UCFL2032**	Symptomatic	106	1072	X4
**UCFL2037**	Symptomatic	419	<500	R5R8
**UCFL2072**	ND	ND	ND	X4
**UCFL2073**	Asymptomatic	ND	ND	R5
**UCFL2074**	ND	ND	ND	R5X4

^a^ Biotype according to co-receptor usage; R5, usage of CCR5; X4, usage of CXCR4; R5X4, usage of CCR5 and CXCR4; R5R8, usage of CCR5 and CCR8; R5X4R8, usage of CCR5, CXCR4, and CCR8; ND, not determined.

**Table 5 viruses-15-01030-t005:** Summary of qualitative results of cis-infection of CD4^+^ T lymphocytes, macrophages, and dendritic cells (both immature and mature) by HIV-1 and HIV-2 primary isolates (data summarized from Figure 1 and Figure 2).

	Isolate	Mø ^a^	ImDC ^b^	Mdc ^c^	TCD4+ ^d^	Biotype
HIV-1	UCFL1014	−	−	−	+	R5X4
	UCFL1026	−	−	−	+	R5X4
	UCFL1028	−	−	−	+	R5X4
	UCFL1029	−	−	−	-	R5X4
	UCFL1032	+	+	−	+	X4
	UCFL1033	−	−	−	+	R5
	UCFL1034	−	−	−	+	R5
HIV-2	ALI	+	+	+	+	R5
	UCFL2017	−	−	−	+	R5X4R8
	UCFL2032	−	−	−	+	X4
	UCFL2037	−	−	−	+	R5R8
	UCFL2072	+	−	−	+	X4
	UCFL2073	+	+	+	+	R5
	UCFL2074	+	−	−	+	R5X4

^a^ Mø, macrophages; ^b^ imDC, immature dendritic cells; ^c^ mDC; mature dendritic cells; ^d^ TCD4+, CD4+ T lymphocytes. A “+” indicates a productive infection, while a “−“ indicates a non-productive infection.

**Table 6 viruses-15-01030-t006:** Summary of the results of the integration of proviral DNA into the genome of cells exposed to different HIV-1 primary isolates and obtained from the nested PCR assay.

Isolate	Mø ^a^	imDC ^b^	mDC ^c^	TCD4+ ^d^
UCFL1014	+	+	+	+
UCFL1026	+	+	+	+
UCFL1028	+	+	+	+
UCFL1029	+	+	+	+
UCFL1032	+	+	+	+
UCFL1033	+	+	+	+
UCFL1034	+	+	+	+

^a^ Mø, macrophages; ^b^ imDC, immature dendritic cells; ^c^ mDC; mature dendritic cells; ^d^ TCD4+, CD4+ T lymphocytes. A “+” indicates successful amplification after nested PCR.

**Table 7 viruses-15-01030-t007:** Summary of the results of the integration of proviral DNA into the genome of cells exposed to different HIV-2 primary isolates and obtained from the nested PCR assay.

Isolate	Mø ^a^	imDC ^b^	mDC ^c^	TCD4+ ^d^
ALI	+	+	+	+
UCFL2017	+	+	+	+
UCFL2032	+	+	+	+
UCFL2037	+	−	−	+
UCFL2072	+	+	+	+
UCFL2073	+	+	+	+
UCFL2074	+	+	+	+

^a^ Mø, macrophages; ^b^ imDC, immature dendritic cells; ^c^ mDC; mature dendritic cells; ^d^ TCD4+, CD4+ 0T lymphocytes. A “+” indicates successful amplification after nested PCR, and a “−“ indicates that no PCR amplicon was obtained.

**Table 8 viruses-15-01030-t008:** Qualitative summary of cis-infection and integration of proviral DNA into the genome of CD4^+^ T lymphocytes, macrophages, and dendritic cells by HIV-2 primary isolates.

Isolate	RT Activity				Integration of Proviral DNA			
	Mø ^a^	imDC ^b^	mDC ^c^	TCD4+ ^d^	Mø ^a^	imDC ^b^	mDC ^c^	TCD4+ ^d^
UCFL1034 ^e^	-	-	-	+	+	+	+	+
ALI	+	+	+	+	+	+	+	+
UCFL2017	−	−	−	+	+	+	+	+
UCFL2032	−	−	−	+	+	+	+	+
UCFL2073	+	+	+	+	+	+	+	+

^a^ Mø, macrophages; ^b^ imDC, immature dendritic cells; ^c^ mDC; mature dendritic cells; ^d^ TCD4+, CD4+ T lymphocytes; ^e^ HIV-1Ba-L (UCFL1034) was used as control.

**Table 9 viruses-15-01030-t009:** Summary of qualitative results of CD4+ T lymphocytes trans-infection by Mø, imDCs, and mDCs infected with HIV-2 primary isolates. The raw data of this set of experiments are included in Appendix A.

Cells	Conditions	HIV-1_UCFL1034_ ^d^		HIV-2_ALI_		HIV-2_UCFL2017_		HIV-2_UCFL2073_		HIV-2_UCFL2032_	
		Donor: Target Cell		Donor: Target Cell		Donor: Target Cell		Donor: Target Cell		Donor: Target Cell	
		1:1	1:4	1:1	1:4	1:1	1:4	1:1	1:4	1:1	1:4
**Mø ^a^**	Cell–cell contact	++	++	-	-	++	+	++	+	+	+
	Transwell separation	+	++	-	-	-	+	-	++	-	-
	Separation after 15 h	++	++	-	-	++	++	++	++	-	-
**imDC ^b^**	Cell–cell contact	++	++	++	++	-	+	++	+	-	-
	Transwell separation	-	+	-	++	-	-	-	+	-	-
	Separation after 15 h	++	++	++	++	-	+	-	-	-	+
**mDC ^c^**	Cell–cell contact	++	++	++	++	-	++	++	+	+	+
	Transwell separation	-	-	-	-	++	-	++	++	-	+
	Separation after 15 h	++	++	++	++	-	-	++	++	-	-

Peak RT concentration was measured in culture supernatants during a 7-day period after virus inoculation: −, peak RT concentration <10 pg/mL; +, peak RT concentration between 10 and 100 pg/mL; ++, peak RT concentration between 101 and 1000 pg/mL. ^a^ Mø, macrophages; ^b^ imDC, immature dendritic cells; ^c^ mDC, mature dendritic cells; ^d^ HIV-1Ba-L reference strain (UCFL1034) was used as control.

## Data Availability

The original contributions presented in the study are included in the article/Appendix A. Further inquiries can be directed to the corresponding author.

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
