# Peer review of "Cell-to-Cell Transmission of HIV-1 and HIV-2 from Infected Macrophages and Dendritic Cells to CD4+ T Lymphocytes"

_viruses, 2023, doi:10.3390/v15051030_

Round 1

Reviewer 1 Report (Previous Reviewer 3)

This manuscript by Calado et al. examines HIV cell-to-cell transmission from infected macrophages and dendritic cells to CD4+ T lymphocytes. The authors use HIV-1 and HIV-2 primary isolates with different phenotypes to analyze the efficiency of HIV-1 and HIV-2 transmission from infected dendritic cells and macrophages to TCD4+. They conclude infected Mø and DCs spread the virus to TCD4 through cell-to-cell transmission.

The authors didn’t response to my comments on the last version or they didn’t have a chance to response. This makes me hard to follow how they improve their manuscript. I still have some concerns about the experiments.
1. In figures 1 and 2, compared with the last version, did the authors completely re-perform the experiments? For the trans-infection assay, did the authors wash the producer cells with PBS remove unabsorbed viral particles?

2. can the authors explain more about the assay of “Separation after 15h”? Is that separation 15 h coculture in trans-well or in cell-to-cell contract condition.

3. The authors added the “Statistical analysis” section in this new version, however they still need to state if the experiments were repeated independently. Otherwise, the technical repeats are not suitable for any statistical analysis.

Author Response

Response to Reviewer 1 Comments

Point 1: The authors didn’t response to my comments on the last version or they didn’t have a chance to response. This makes me hard to follow how they improve their manuscript. I still have some concerns about the experiments.

Response: In fact, we have answered all the questions raised by the reviewers as stated in the cover letter to the Editor during the resubmission process. In addition, where necessary, further experiments were performed and some details were added to improve the presentation of the results. Since this was a resubmission, I assume that this reviewer didn't have a chance to see our responses.

Point 2: In figures 1 and 2, compared with the last version, did the authors completely re-perform the experiments? For the trans-infection assay, did the authors wash the producer cells with PBS remove unabsorbed viral particles?

Response: The experiments that yielded the results shown in Figures 1 and 2 have not been repeated, although some details that were missing in the original version are now included. For example, we have changed the Y-axes to more clearly show the range of results obtained. Another detail missing from the original submission was the explicit mention of how we wash the cells after the initial inoculum in trans-infection as well as in cis-infection. The resubmitted version clearly states this (lines 145 and 174-175, respectively), among many other statements to avoid misinterpretation throughout the text.

Point 3: Can the authors explain more about the assay of “Separation after 15h”? Is that separation 15 h coculture in trans-well or in cell-to-cell contract condition.

Response: The separation of target and donor cells was done in the coculture assay when donor and target cells where in contact. We wanted to see if direct contact between donor and target cells was a strict requirement to see infection of the target cells, and we wanted to compare this to a culture condition where the cells could not contact each other (the transwell condition). We have elaborated on the rationale underlying this set of experiments in both the MM and Results sections (lines 177-179 and 440-443, respectively).

Point 4: The authors added the “Statistical analysis” section in this new version, however they still need to state if the experiments were repeated independently. Otherwise, the technical repeats are not suitable for any statistical analysis.

Response: Thank you for reminding us of this... this information was missing and is now explicitly stated what was done regarding the replication of all experiments presented: triplicates in each set of three independent experiments. This was included in MM and Results section (lines 191-192 and 351-352 respectively).

Reviewer 2 Report (New Reviewer)

The manuscript by Calado, et al describes the contribution and efficiency of cis- (cell free) or trans- (cell-associated) infection of HIV-1 and HIV-2 isolates between susceptible cell types: macrophages, dendritic cells, and CD4+ T lymphocytes.  While important to understand transmission events, especially in primary isolates, the manuscript would benefit from a variety of clarifications:

Major Comments:

1.     Lines 87-91.  The last paragraph of the introduction would benefit from mention of specific results/conclusions obtained from this study.

2.     Lines 94-96.  Please explain why PBMCs from three different donors were pooled together.  Are the authors concerned about mixed leukocyte responses?  The value of repeated measure from independent donors is lost.  Regardless, the number of donors should be indicated for each set of data.

3.     Lines 123-124.  Please indicate the source of the HIV-1 and HIV-2 PBMCs/isolates.  The M&M section only indicates the source of the PBMCs from healthy donors.

4.     Liens 125-126.  Please confirm that reverse transcriptase (RT) activity was measured in culture supernatants by ELISA.  The data shown indicate a concentration of RT on the y-axis in pg/ml.  Therefore, the ELISAs are likely measuring the amount of RT present as a proxy for the amount of virus present, as opposed to RT activity.  Please clarify throughout the manuscript where “RT activity” is referenced.

5.     Lines 132-134.  Please clarify what is meant by the fact tha trains were selected from a library.  This section also states that virus isolates were grown out from infected PBMCs.  Were these selected from a library of patients, PBMCs, or virus stocks.

6.     Table 3 and 4: Please indicate the units for CD4+ lymphocytes and viral load

7.     Please provide justification for why HIV-2 isolates were used for trans-infections and not HIV-1.

8.     It is indicated that virus strain UCFL1034 is actually HIV-1 BaL.  Is this considered a HIV-1 primary isolate or a lab adapted strain?

9.     What is the minimum concentration of virus measured to be considered positive for a spreading infection?  For example, UCFL2073 was indicated to replicate in macrophages, but the data shown do not appear convincing.

10.  The manuscript may benefit from quantification of viral mRNA as opposed to just integrated provirus.

11.  The phenotype of the macrophages, DCs, and CD4+ T-cells becomes important when conclusions are drawn about receptor and co-receptor usage.  The expression of these receptors should be quantified on these cells in order to make conclusiosns.

12.  Table 9 – the raw data summarized here for the trans-infection assays should be shown.  Comparisons and individual p-values should be indicated.

13.  Please comment as to the different donor:target cell ratios and what results yielded from those experiments.

14.  What do the error bars represent on the figures?

Minor Comments:

1.     Table 9.  This title needs to include HIV-1 and HIV-2 isolates.

2.     Throughout the manuscript, “retrotranscription” should be changed to “reverse transcription”

Author Response

Response to Reviewer 2 Comments

Point 1: Lines 87-91.  The last paragraph of the introduction would benefit from mention of specific results/conclusions obtained from this study.  

Response: This is a point that was missing in the resubmitted version of the manuscript. We have now added a few lines to this effect (lines 89-92).

Point 2: Lines 94-96.  Please explain why PBMCs from three different donors were pooled together.  Are the authors concerned about mixed leukocyte responses?  The value of repeated measure from independent donors is lost.  Regardless, the number of donors should be indicated for each set of data.

Response: Our protocol for decades has been to pool the uninfected cells in one batch. Therefore, we use different Buffy coats as the source of PBMCs. This avoids individual biases such as increased or decreased susceptibility to infection, variations in cell activation, genetic makeup, etc. The cells were always from 3 different Buffy coats as mentioned in the MM section.

Point 3: Lines 123-124.  Please indicate the source of the HIV-1 and HIV-2 PBMCs/isolates.  The M&M section only indicates the source of the PBMCs from healthy donors.

Response: Thank you for this comment. The HIV isolates were obtained from cocultures between PBMC from uninfected patients (buffy coats) and PBMC from blood samples from HIV-1 and HIV-2 infected patients. This was done in a previously published paper as now mentioned in the MM section (lines 129-130).

Point 4: Lines 125-126.  Please confirm that reverse transcriptase (RT) activity was measured in culture supernatants by ELISA.  The data shown indicate a concentration of RT on the y-axis in pg/ml.  Therefore, the ELISAs are likely measuring the amount of RT present as a proxy for the amount of virus present, as opposed to RT activity.  Please clarify throughout the manuscript where “RT activity” is referenced.

Response: Absolutely correct. The ELISA we used to quantify RT activity is based on an in vitro system where the result is a proxy for the concentration of RT. We have now corrected this throughout the manuscript. However, we believe that the assumption of RT activity based on this relationship is legitimate and does not compromise the results.

Point 5: Lines 132-134.  Please clarify what is meant by the fact tha trains were selected from a library.  This section also states that virus isolates were grown out from infected PBMCs.  Were these selected from a library of patients, PBMCs, or virus stocks.

Response: We used viruses that are part of our extensive virus library, and these HIV isolates were isolated, as mentioned above, by coculturing PBMC from infected patients with PBMC from blood donors (without HIV infection). Thus, what we used were viral stocks propagated in PBMCs as referred to in a previous paper. We have clarified this potential misinterpretation in the manuscript (lines 129-130).

Point 6: Table 3 and 4: Please indicate the units for CD4+ lymphocytes and viral load.

Response: We have now included the units of both CD4+ lymphocyte count and viral load.

Point 7: Please provide justification for why HIV-2 isolates were used for trans-infections and not HIV-1.

Response: The main reason why we choose to used HIV-2 isolates for trans-infection assays was because we observed, in cis infection assay, that HIV-2 isolates were a more heterogeneous group regarding their ability to infect Mø, imDCs and mDCs. Furthermore, the incidence of HIV-2 infection remains high in Europe, particularly in our country Portugal, so it is of paramount importance more studies for a better understanding of the mechanisms underlying HIV-2 transmission. Moreover, HIV-2 is an underappreciated model for study the retrovirus-human interactions that we always aimed to explore throughout our research. We explain these in lines 413-417.

Point 8: It is indicated that virus strain UCFL1034 is actually HIV-1 BaL.  Is this considered a HIV-1 primary isolate or a lab adapted strain?

Response: This is an important comment that we should have clarified. The HIV-1 Ba-L is a widely used Mø-tropic prototype strain. We used it as a control based on that phenotype. So it is not a primary isolate, it is a reference strain that was obtained from a regent repository.

Point 9: What is the minimum concentration of virus measured to be considered positive for a spreading infection?  For example, UCFL2073 was indicated to replicate in macrophages, but the data shown do not appear convincing.

Response: Inadvertently we did not include this information. The minimum concentration of virus measured that was considered positive was 10 pg/ml. We have now added this data in the legend of Figure 1 and Figure 2. We also clarified some results in lines 345-354.

Point 10: The manuscript may benefit from quantification of viral mRNA as opposed to just integrated provirus.

Response: We agree with this observation. However, our goal was to see if there was blockade at the level of viral entry and retrotranscription. We all agree that the detection of chromosomally integrated proviral DNA is an accurate way to observe this. We are not necessarily interested in whether this leads to proviral transcription or translation, but only in the ability of the virus to enter, to retrotranscribe its RNA, and whether the resulting dsDNA was integrated or not.

Point 11: The phenotype of the macrophages, DCs, and CD4+ T-cells becomes important when conclusions are drawn about receptor and co-receptor usage.  The expression of these receptors should be quantified on these cells in order to make conclusions.

Response: We fully agree with this observation. In fact, this is the reason why all viruses used were previously characterized for their ability to use different coreceptors together with CD4 (Calado et al. Virology, 2010). Note, however, that we are comparing viruses using the same batch of cells. Therefore, if there were large variations in cell coreceptor expression, this would affect all results, which was not the case.

Point 12: Table 9 – the raw data summarized here for the trans-infection assays should be shown.  Comparisons and individual p-values should be indicated.

Response: Initially we did not include the raw data of trans-infection assays because of the substantial amount of results we had obtained from this part of the work. We thought it was more clearly to present the data summarized as we did in Table 9. However, we understand the importance of showing the raw data so included this as Supplemental Results and referred this fact in the legend of Table 9.

Point 13: Please comment as to the different donor:target cell ratios and what results yielded from those experiments.

Response: Thank you for this important point. The rationale for using different donor:target cell ratios (1:1 and 1:4) was to see if the presence of an increasing amount of target cells could somehow influence the efficiency with which virus was transferred. The results show that in several cases an increase in virus transfer was observed when we used a 1:4 ratio. We have not discussed this further because it was not a clear and general pattern and we believe it may be a strain-specific effect. However, we have now included a brief discussion of these results (Lines 478-481).

Point 14: What do the error bars represent on the figures?

Response: Thank you for reminding that we did not explain what the error bars represents on the figures. We added the explanation on the figures legends.

Minor Comments:

Point 15: Table 9.  This title needs to include HIV-1 and HIV-2 isolates.

Response: This table includes results from HIV-2 assays only. The HIV-1Ba-L (UCFL1034) was used only as a control.

Point 16: Throughout the manuscript, “retrotranscription” should be changed to “reverse transcription”

Response: This change was made throughout the manuscript.

Round 2

Reviewer 1 Report (Previous Reviewer 3)

The requested changes to the figures are sufficient. As a result, the quality of the paper is substantially improved.

Reviewer 2 Report (New Reviewer)

The authors have satisfied all comments.

This manuscript is a resubmission of an earlier submission. The following is a list of the peer review reports and author responses from that submission.

Round 1

Reviewer 1 Report

Attached

Reviewer 2 Report

Overall: Marta Calado et al. provide an article about the transmission from HIV-1 and HIV-2 infected-macrophage/dendritic cell to CD4+ T cell in in vitro culture. This article can be an important input about best understanding of spread of HIV infection and the HIV reservoir establishment. However, some points must be reinforced to claim this objective.

Major comments:

1.     The authors show only one time the data exhibit (Figure 2 and 3). No data was displayed for Table 6, 7, 8, 9. The summaries are not enough the authors must show and present the data obtained from the experiment. Thus, in general this is difficult to discuss about the result because the data is never exposed.

2.     The correlations mentioned is the result must be shown as table or graph (R and pvalue).

3.     Nothing is explained about the day post-infection in the Table 5. If the table presents the day 12 post-infection, and if the table show HIV production at this day; your data from figures 2 and 3 do not show the HIV production from imDC :UCFL1032/ALI/UCFL2073, Mø: ALI/ UCFL2073/ UCFL2074, mDC: ALI/ UCFL2073, TCD4+: UCFL1029/ UCFL2017/ UCFL2032/ UCFL2037, like exposed in the table 5.

4.     In general, the manuscript and specially the result section must be clarified about the data exposed.

5.     What is the initial MOI selected by the authors to infect the cells in cis-infected?

6.     Why are there no data in figure 2 and 3 at day 12 post-infection for patients UCFL2072 and UCFL1032?

7.     This is also important to discuss about the difference between transwell condition and separation condition could be due to the cytokines production and free virus can pass through the transwell (0.4um).

8.     Unfortunately for human study, the authors do not have a lot of patients. If this is possible I recommend to add more patients in this study.

Minor comments:

1.     In the abstract, the result section is not clear about what you observe in the article.

2.     One more space at the line 66 before the period.

3.     Cis-infection is not defined in introduction compared to trans-infection.

4.     Do the authors check what does the proportion of competent virus produce by the coculture?

5.     The CXCR4 and CCR8 virus tropism must be mentioned in the introduction.

Reviewer 3 Report

This manuscript by Calado et al. examines HIV cell-to-cell transmission from infected macrophages and dendritic cells to CD4+ T lymphocytes. The authors use HIV-1 and HIV-2 primary isolates with different phenotypes to analyze the efficiency of HIV-1 and HIV-2 transmission from infected dendritic cells and macrophages to TCD4+. They conclude infected Mø and DCs spread the virus to TCD4 through cell-to-cell transmission.

The authors have performed a substantial amount of work and my impression is that they have tried very hard to present the results clearly, but at places this breaks down due to the somewhat convoluted system and the quite complex results that they obtain.

Discrepancies in the results are not commented upon, and they need to be carefully addressed. While many of the results are interesting, cell-to-cell transmission from HIV-1 transfer from DCs and macrophages has been studies by many groups, and some important controls and experimental points are missing.

The following points should be addressed.

1. In Figure 1, how do the authors define “productive infection”? e.g. In line 189-190, how do the authors conclude Mø infection by UCFL2073, ALI, UCFL2074? I only saw a small increase of RT activity in the cells infected with UCFL2072. How do the authors define the time point “T 0” in this figure? The authors need to carefully explain the whole figure.

2. Table 6, instead of showing the positive or negative of DNA integration, can authors show these data quantitatively? What is the negative control of this assay?

3, in line 158, “the virus-cell mixture was cocultured with autologous activated CD4+ T lymphocytes”. Why do the authors add virus-cell mixture to the target cells but not only infected producer cells? This means the target cells can be infected by cell-free virus.

4, in line 160, can the authors explain more about the assay of “Separation after 15h”? e.g. how to separate the donor from the target cells 15h after the infection.

5, in line 270, the authors state that “UCFL2073 replication only occurred when the donor and target cells were in contact or when they were separated by a transwell”. From what I understand, the cell-free virus infection is measured by the transwell assay. The infection in transwell means cell-free virus infection happened here. Why do the authors conclude that” therefore, the production of viral particles is probably induced by the co-culture of donor and target cells”?

6, The experiments in this manuscript were only performed once statistically. The authors need to repeat their results independently.